# The Validity of Machine Learning Procedures in Orthodontics: What Is Still Missing?

**DOI:** 10.3390/jpm12060957

**Published:** 2022-06-11

**Authors:** Pietro Auconi, Tommaso Gili, Silvia Capuani, Matteo Saccucci, Guido Caldarelli, Antonella Polimeni, Gabriele Di Carlo

**Affiliations:** 1Private Practice of Orthodontics, 00012 Rome, Italy; pietroauc@gmail.com; 2Networks Unit, IMT School for Advanced Studies Lucca, Piazza San Francesco 19, 55100 Lucca, Italy; 3ISC CNR, Department of Physics, University of Rome “Sapienza”, P.le Aldo Moro 5, 00185 Rome, Italy; silvia.capuani@isc.cnr.it (S.C.); guido.caldarelli@unive.it (G.C.); 4Department of Oral and Maxillo-Facial Sciences, Sapienza University of Rome, Viale Regina Elena 287a, 00161 Rome, Italy; matteo.saccucci@uniroma1.it (M.S.); antonella.polimeni@uniroma1.it (A.P.); gabriele.dicarlo@uniroma1.it (G.D.C.); 5Department of Molecular Sciences and Nanosystems, Ca’Foscari University of Venice, Via Torino 155, Venezia Mestre, 30172 Venice, Italy; 6ECLT, Ca’ Bottacin, Dorsoduro 3246, 30123 Venice, Italy

**Keywords:** machine learning, artificial intelligence, orthodontics, complexity, prognosis optimization

## Abstract

Artificial intelligence (AI) models and procedures hold remarkable predictive efficiency in the medical domain through their ability to discover hidden, non-obvious clinical patterns in data. However, due to the sparsity, noise, and time-dependency of medical data, AI procedures are raising unprecedented issues related to the mismatch between doctors’ mentalreasoning and the statistical answers provided by algorithms. Electronic systems can reproduce or even amplify noise hidden in the data, especially when the diagnosis of the subjects in the training data set is inaccurate or incomplete. In this paper we describe the conditions that need to be met for AI instruments to be truly useful in the orthodontic domain. We report some examples of computational procedures that are capable of extracting orthodontic knowledge through ever deeper patient representation. To have confidence in these procedures, orthodontic practitioners should recognize the benefits, shortcomings, and unintended consequences of AI models, as algorithms that learn from human decisions likewise learn mistakes and biases.

## 1. Introduction

For thirty years now, AI procedures have proven to be highly effective tools when implemented correctly, allowing one to perceive subtle information and ultimately convert information into actions, from speech recognition to natural language processing, spam filters, fraud detection, and many other applications [1,2,3,4,5,6]. Machine learning (ML), a subcategory of AI, is the method of creating models that perform a specific task without the need to be explicitly programmed by a human for the discovery of intricate correlations within large masses of data [7,8,9,10]. In biomedicine, ML tools have been widely applied in the handling of large numbers of patient micro-variables and in predicting the future outcomes of diseases based on previous data collected regarding similar diseases [11,12,13,14,15,16,17]. Despite its success, ML is a still-emerging technique in medicine, even more so in orthodontics, and many opportunities remain unexplored.

Physicians seem to overlook the fact that machine judgment is, on average, at least as reliable as expert judgments and, in many circumstances, even better [6,9]. In recent years, an excessive tendency to rely on ML systems (“over-reliance”), overdependence, “deskilling”, and even potential desensitization to patient problems have been the most common criticisms related to the use of these models in medicine [18,19,20,21]. Concerns have arisen regarding the possibility of incorporating “dirty” data and spurious correlations based on uncertain clinical interpretations [22]. Some practitioners believe that using computational systems could lead to the establishment of a new, problematic medical empiricism, based not on concrete facts and data relating to patients but rather on interconnections of data, meaning that the most straightforward clinical situations can become complicated and confusing [23]. Is there any truth to these claims? If it is true that automation will not turn us into robots (just as industrialization did not turn us into machines), is it possible that these computational procedures could increase rather than decrease the risk of statistical-clinical misunderstandings? In this paper we discuss the persisting unresolved issues related to ML computational predictive tools applied to orthodontics, which still limit the automatic extraction of valuable, clinically actionable orthodontic knowledge.

## 2. Challenging Interface between Machine Learning Models and Orthodontic Features

Many reports have highlighted the usefulness and potential of predictive electronic systems in clinical and research orthodontics [23,24,25,26,27]. The potential identified in these reports is similar to what has been hypothesized in medical practice—the records of the best clinical decisions made by thousands of professionals must be exploited to optimize patient care [28,29,30]. Although ordinary medical diagnostic approaches are based on the slow, careful recruitment of clinical and laboratory data, on subjects including the causes and effects of clinical phenomena, the significance of symptoms, and so on, the most sophisticated predictive ML implementations learn and store information at great speed, solving complex problems by repeatedly re-examining the data and layering simple concepts onto more complex ones. Translating from daily orthodontics to ML models, an orthodontist might (i) identify hidden craniofacial trends in large datasets of growing patients, (ii) leverage trends to make growth outcome predictions, (iii) compute the probability for each possible growth and treatment outcome, or (iv) clarify the effects of the co-occurrence of skeletal defects and the renormalization phenomena on growth and treatment outcomes [28,29,30,31,32,33,34].

The fundamental requirement of predictive analytics procedures applied to orthodontics concerns the availability of accurate clinical data, with which the machine can gain “domain experience” [35,36]. Computer scientists use what they know how to do (algorithms) to data, the peculiarities of which they do not always understand. ML developers assume that the patient dataset used to train models is uniformly and fully representative of the target patient population. However, in medicine, not all subjects are equal. Some patients give more representative information about their clinical condition than others. Moreover, the reference data of some patients may not be 100% accurate. Thus, the process of weighing higher-quality information against other information is debatable and subjective, so the data quality dimension in orthodontic records remains a cognitively elusive concept [12,16,17]. There are differences between actual craniofacial morphology assessments, as experienced by orthodontists, and their codified representation in a numerical form, which is the case for the data input for any ML algorithm.Computational machines teach orthodontists to map medical phenomena into numerical structures in order to quantify them. Based on associations, algorithms can exploit features that the orthodontist may consider irrelevant to the problem. It is not essential to understand in depth the functioning of various patient characteristicsin order to extract answers. It is necessary to engage with constant change, randomness, and noise in the data relating to orthodontic treatment and look for regularities within the data, rather than for for clinical-logical hypotheses. In other words, one must allow the numbers to speak for themselves (Figure 1).

Concerning the use of ML models, there are problems related to the the steps necessary to move from the observed patient data to the statements concerning future patients who have never been seen before. Machines can capture hierarchical regularities and dependencies in the data to learn complex correlations between input and output features without any inherent representation of causality [10,13,21,22]. ML procedures do not require theoretical bases.Unfortunately, predictions based on data rely on mere correlations without theories and models. A correlation quantifies the statistical relationship between the values of two parameters without clarifying the inner mechanisms of a system. Probabilities based on scarce data are not reliable, but it is not just a question of quantity. The primary source of data for the training of ML orthodontic models is data produced during growth and/or during treatment. The most difficult factor to respond to is the possible shifts between time-series data that may induce temporal drifts, which can cause algorithms to become progressively inaccurate. Craniofacial data continually evolve throughout growth, so the future does not always look like the past. Given these premises, one may suspect that the procedural ML logic applied to orthodontics may pose more than one interpretive problem. However, sufficient evidence has accumulated that ML tools, when applied to orthodontics, perform well [22,23,24,25,26].

When numbers are the only object of interest in diagnostic and prognostic processes, some dehumanization occurs. Consequently, a certain amount of ineradicably intrinsic potential distortion in the interpretation of orthodontic conditions may arise. However, humans are fortunate to possess a fundamental property that the most up-to-date ML systems lack: common sense. Humans can infer the reasons behind processes, identifying abstract similarities and analogies based on only a few observed patients. Machines have facilitated disparate orthodontic clinical situations: diagnostic assessments; prognostic predictions; and the identification of salient features in growing patients at risk of skeletal imbalance, poor response to treatment, maxillofacial tumours, cysts, periapical abscesses, etc. [27,28,29,30]. The accurate localization of cephalometric landmarks using ML tools has led to a mitigation of the problem of interpersonal variation in landmark tracing and related errors in diagnosis and treatment planning [24]. The application of photography-based systems to assess jaw disharmonies (responsible for masticatory dysfunctions and apnea syndrome) and to establish the need for extractions in cases of tooth crowding and protrusion, are additional crucial steps for the successful application of ML-supported decision procedures [25,26] (see Appendix A).

## 3. How Can Orthodontic Input Be Incorporated into the Machine Learning Process?

During the growth process, the clinical and cephalometric data used to feed ML machines have inherent randomness related to massively parallel processes of skeletal plasticity, which propagate through algorithms with an unavoidable degree of inaccuracy. The stochastic processes that cause a developmental trait to deviate from its expected path [20], also known as developmental noise, are an inherent part of craniofacial development and remodelling. Significant variations at the organ and whole organism level are related to the stochasticity of random intermolecular collisions, gene fluctuations, signal transduction factors, chromatin structure, DNA methylation state, morphogenetic cytoskeleton dynamics, bone translations, and other factors [20,21]. The complex pathobiology of craniofacial growth recalls a well-known saying among data scientists: all data is dirty. Nevertheless, the hypothesis offered by electronic systems is that the combination of multiple subtle aspects and a sequence of non-linear data transformations can be performed to extract both clinical and subclinical patient nuances (Figure 2), covertly containing the answer to a given problem.

In the unfolding of clinical reasoning, physicians make diagnostic errors 5%–15% of the time, depending on their speciality [16]. Two to four pieces of clinical information are sufficient to generate diagnostic hypotheses through intuition. The errors are related to the fast closure of the diagnostic process, as well as the tendency not to consider alternative views to the first diagnosis (“anchoring bias”), the tendency to consider diagnoses that are easy to remember (“availability bias”), and the tendency to include only confirmatory data for the initial diagnosis while ignoring contradictory data (“confirmatory bias”) [13,17]. Conversely, errors for machines mainly occur during the learning step. Since machines do not have the capability for intuition, the most common cause of errors lies in the poor quality of training data, such as irrelevant features, spurious associations, false assumptions, inappropriate patient attributions, and indications that are unable to represent the patient’s clinical background [9,10,11,12]. Computational models define their reality and use it to justify their results and make predictions. However, living organisms cannot be reduced to a set of mathematical equations suitable for describing an elementary mechanism; the internal parts are not endowed with the statistical homogeneity that would allow the application of probability theory. Craniofacial imbalance constitutes a repository of physical order in which a large amount of information is concentrated. Patients’ unequal developmental probabilities are due to morphological constraints, competition/collaboration strategies of skeletal elements, emergent phenomena, bone translations, and more. Patients with severe facial imbalance escape dento-alveolar renormalization systems since they tend to maintain disharmony over time (Figure 3). Some developmental properties directionally constrain the possible path of evolution, defining the limits of the possible craniofacial variations associated with that specific initial morphology (“canalization”) [13]. Intuitively, the numerical transposition of these concepts is somewhat problematic. Algorithmic decisions are expressed in the form of rigid, not-fuzzy binary classifications (spam-not-spam, dog-cat, etc.). Making prognostic clinical predictions means identifying the presence of unfavorable factors when they have not yet occurred. To obtain a satisfying computer-assisted predictive ability, the operator must provide the machine with a series of expressive examples of the condition to be detected, i.e., examples of patients with signs and symptoms typical of the disease, which can easily be differentiated from healthy, symptom-free patients [35,36,37,38]. In the orthodontic scenario, in the same patient, shaded morphological/radiographic features of malocclusion may coexist with typical craniofacial characteristics or even with signs of a different malocclusion. There is no such vagueness in mathematical language. In mathematical language, everything is precise. Machines tend to complete information when only part of the system is known. Each orthodontic patient contains a different amount of hidden data and latent variables not expressed in numerical format (the “hidden half”) [20]. All of these affect the outcome, so two similar patients can carry a very different facial growth potential and potential responses to treatment related to different inherent amounts of developmental noise. Strains related to the imbalance between ideal prognostic models and the everyday fuzzy orthodontic reality are called “misdiagnoses” and “wrong prognoses” by orthodontists and “residuals” by statisticians (see Appendix B).

## 4. Tell Me What You Have Understood about This Patient

Practitioners generally trust their subjective intuition more than the answer provided by an algorithm [2,3,39,40]. Humans are poor at making probabilistic decisions based on partial information and cannot even precisely calculate how data interfere with each other [31,41]. As already mentioned, in a patient dataset, some components (for instance, dentoalveolar adaptive remodelling) can remain below the threshold of perception of ML tools [42,43,44,45,46,47,48,49,50,51,52,53,54,55]. Some features may be irrelevant, missing, or redundant. The most up-to-date deep artificial neural networks do not require any additional pre-processing; they automatically cut out uninteresting correlations between parameters to build up a meaningful subset of data (Figure 4).

This procedure allows for the comparison of the results expected by an expert, based on experience, and what is discovered by means of computational rules. One of the fascinating elements of ML algorithms lies in their ability to attribute the degree of reliability of each prediction to a self-validation process [1,2,3]. The ability of machines to give proper weight to the various patient factors involved in the prediction is much larger than that of humans. Paradoxically, to optimize predictions and to generalize to as many patients as possible, the software logic requires individual patient specificities to be flattened out (the “regularization” procedure) [17,18,51]. A system that is too smart in the diagnostic process ends up focusing too much on individualized patient information and has difficulties in diagnosing new patients with only slightly different characteristics (the “overfitting” phenomenon) [1,2,3]. To facilitate the generalization of the model performance, sometimes there is a need to inflate the data set with confounding noise. There is also the possibility of implementing procedures of data augmentation, as ML systems can create additional “synthetic patients” to improve the accuracy of forecasts [20,52,53,54,55].

Without proper professional orthodontic supervision, the science that has allowed us to refine the patient description through feature engineering and feature selection risk may lead us to implement naïve approaches and to base decisions upon elementary clinical-technological, over-purified versions of patients. An example of the difficult balance between patient specificity, patient context, and computational answers is offered by the (sometimes too ingenuous) treatment programs underlying dental aligners. To achieve the desired outcome, a good dental alignment program, in theory, should be able to incorporate the interactions between dental movements and facial aesthetics and account for the co-occurrence of different patient characteristics, including skeletal and functional constraints, atypical swallowing, mouth breathing, and many others [24]. In the age of big data in biomedicine, it is becoming less and less possible to know in advance the direction and nature of calculations based on collective data [32,33,34,37,38,39,40,41,42,43,44,45,46]. Currently, fully automated methods for model selection and automatic parameter optimization are available, such as AutoML, neural architecture search, differentiable architecture search, reinforced learning, and many others [54,55]. These procedures allow the discovery of data architectures that are far more complicated than those which humans may think of trying. As there is an apparent difference between patient recognition and genuine comprehension, the orthodontist must attempt to integrate computational responses with his cognitive cause-and-effect system carefully. Often, the advice is to broaden the patient sample to better frame the system’s structure. The amount of data does not allow the consideration of fundamental questions regarding the validity of constructs such as the question of whether the patient traits are stable and comparable across patients and over time. When searching a more extensive biomedical database, it is easy to find a pattern that seems interesting, even when it is not actually relevant.Each random dataset observed over time can determine any pattern [17]. Despite the element of predictability that is missing (not expressed in the available data), relying on the algorithmic outcome prediction means trusting the ability of computational abstractions to nevertheless understand the patient by probing deeply and recursively into both visible and latent attributes. The computer-aided orthodontic operator hopes to overcome the prognostic uncertainty through repeated “deep” situational data abstractions, applying a more significant number of patients and a greater number of layers of computation.

## 5. A Matter of Trust

Human memory is an active process, based on encoding, storing, and retrieving previously acquired information [35,42]. At the chairside, orthodontists make reasoned decisions based on the logic of biomechanics and a somewhat schematic taxonomy of malocclusions. Their cognitive statistics (experience) highlight the underlying prevailing clinical trends for each patient and elements that are not very or not at all “datable”, such as cultural and family aspects, compliance, and others [39]. ML statistics help orthodontists to highlight the outcome probabilities and the probabilities of escaping these trends. To disseminate the best practices and to enable researchers and practitioners to trust ML procedures, they first need to understand the bases underlying the algorithmic decisions and predictions. This would require a comprehension, at least in principle, of differences in numerical and orthodontic formalisms within the inscrutable hidden “black box” of algorithms [56]. Although the nature of ML optimization is purely mathematical, craniofacial feature optimization during growth is, above all, a matter of adaptation [57,58,59]. The best possible clinical-digital model may include neither the past nor the present, but only a situation calculated at every moment. When prognostic processes are conducted across both technological and morphological boundaries, new orthodontic theories could be derived through the pure power of technology [58,59,60,61]. Machines must be understandable and acceptable, even though the understandability of the algorithmic answers is often inversely proportional to the transparency and the complexity of the predictive models [46] (Figure 5). Computational models attempt to organize thoughts. Despite the necessary refinements that have been, when applied to orthodontics, these procedures have been proven to improve the professional skills of orthodontists and will soon do so even more effectively. For the benefits of a more productive man-machine operational coupling, future research should focus on a new form of digital ecology. Specifically, better interactive methods are needed for dealing with residual algorithmic standardization issues, better guidelines of algorithmic procedures, and better governance of the processes involved in the creation, validation, and updating of predictive orthodontic models.

## 6. Conclusions

Orthodontics is characterized by prognostic uncertainty, with a strong influence of factors that are not easy to model. Therefore, reliable computerized predictive tools and procedures could be particularly welcome, even from a cautionary and medical-legal point of view. The use ML will not be able to replace orthodontists in the coming years. It will be used in cooperation with orthodontists to enhance their abilities and clinical sagacity. The significance of ML results is required to be verified repeatedly by orthodontists, patients, and computer scientists, using a stable and shared interpretative framework, in order for this technique to be more extensively applied in research and in clinical orthodontic practice.

## Figures and Tables

**Figure 1 jpm-12-00957-f001:**
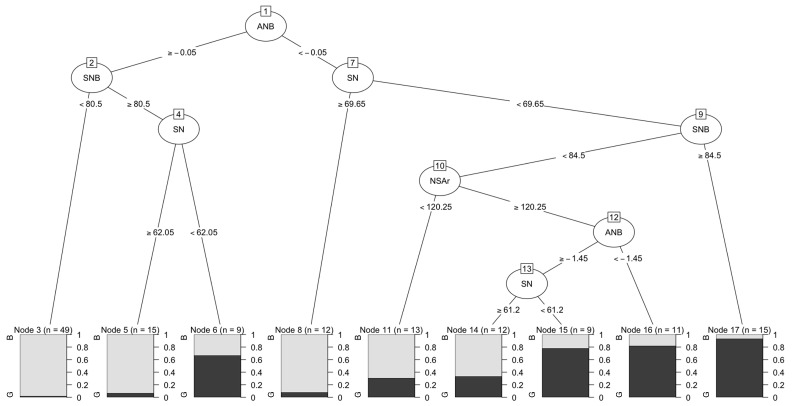
Classification trees are predictive models that can be used to decompose a problem into increasingly simple subcomponents. A tree is composed of branching processes that emerge from a series of binary selections (which are set to be larger or smaller than a reference). Tree algorithms learn through repeated exposure to clinical cases (“examples”). For classification purposes, a tree is built by repeatedly dividing the observations (e.g., cephalometric features as independent input features) into subsets that are as homogeneous as possible in relation to the dependent variable (the label). For learning to occur, data used for training must be labeled. In the example provided, the time progressionof cephalometric data from 80 class III male and female growing subjects (aged from 7 to 14 years) was associated with a label indicating good (improving) or bad (worsening) craniofacial growth. The initial ANB angle reference (−0.05 degrees) was chosen to start the branching process, including other cephalometric characteristics (each associated with a specific reference). In the end, it was possible to establish whether the whole configuration was associated with bad (B) or good (G) growth. This simple procedure may constitute a prognostic aid for the orthodontic operator in communicating risk to parents. The symbolic learning related to classification trees is probably the most expressive procedure for medical data analysis when interpretability is desired. Trees were produced using the R package “tree” v1.0-37. ANB angle (degrees): measure of the relative position of the maxilla to the mandible; SNB angle (degrees): measure of the angle between the sella/nasion plane and nasion/B plane; NSAr angle (degree): measure of the angle between the anterior and posterior cranial base; SN (mm): antero-posterior length of the cranial base. Patient datawere kindly offered by professors Lorenzo Franchi and James A. McNamara Jr.

**Figure 2 jpm-12-00957-f002:**
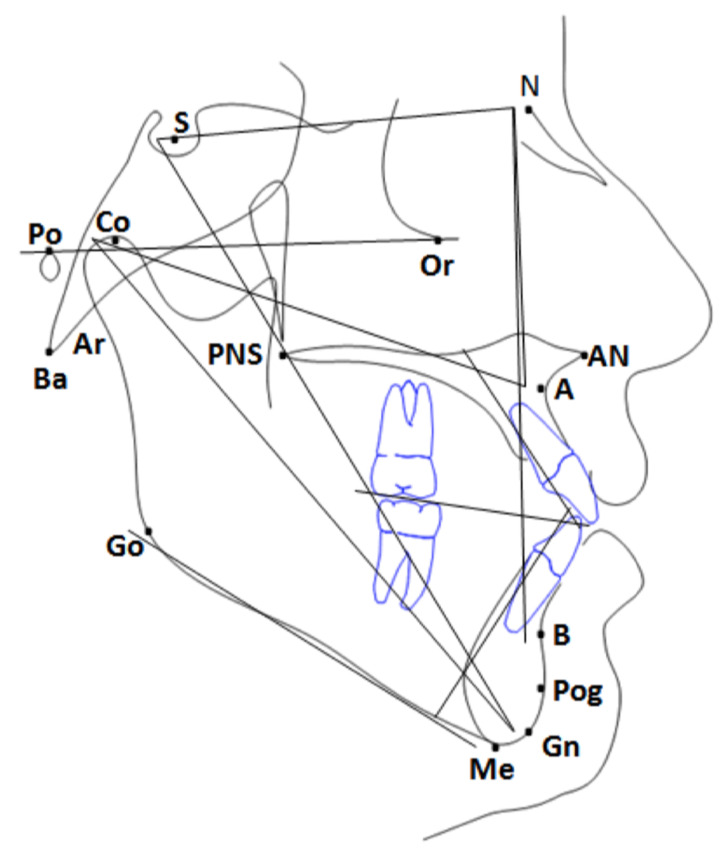
Cephalometric angular and linear measures.

**Figure 3 jpm-12-00957-f003:**
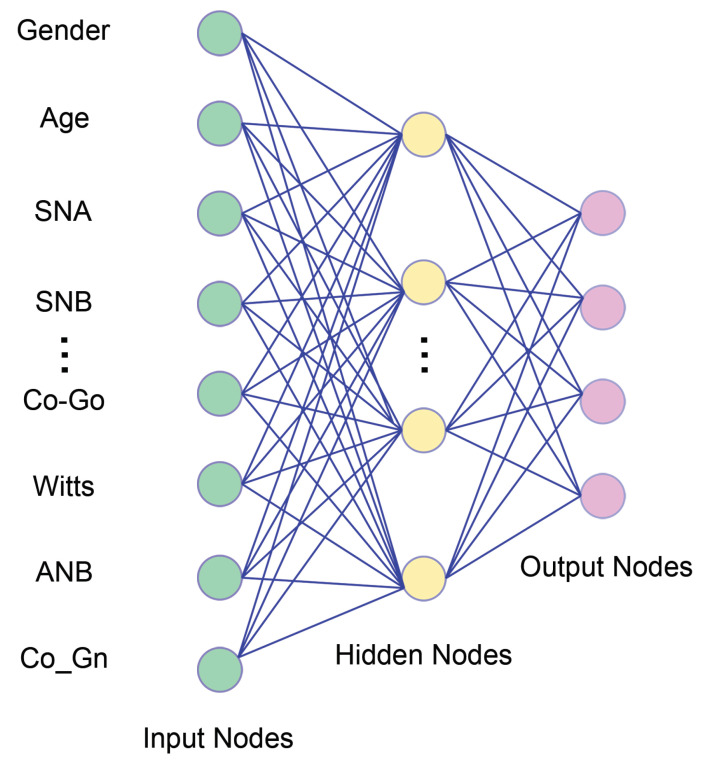
A simplified Neural Network. Artificial neural network (ANN) models can “learn” from the data without any pre-specified rules and can focus mathematically on predictive performance. ANNs take the raw data at the lower (input) layer and transform them into an increasingly “abstract” representation of the characteristics. ANNs are flexible and versatile tools. A few assumptions are required about the normal distribution of errors, correlations among variables, and linear relationships among variables. They are highly applicable for any real-world situation but require many attributes and observations. The difficulties in ANN research applied to orthodontics come in many different forms. The most important contribution is the lack of a uniform feature standards in building ANN models. The second primary reason is that ANNs have fewer assumptions and many more options in the modelling process, which opens up several possibilities for their inappropriate use and applications. Deep learning is a type of ANN procedure carrying multiple Although node layers. Each layer learns the representation of data by abstracting data in many ways. While traditional statistical techniques require transforming raw data (feature engineering) to represent the problem and make predictions, deep learning algorithms achieve this automatically, using more and more abstract levels of representation, encapsulating highly complicated functions in the process.

**Figure 4 jpm-12-00957-f004:**
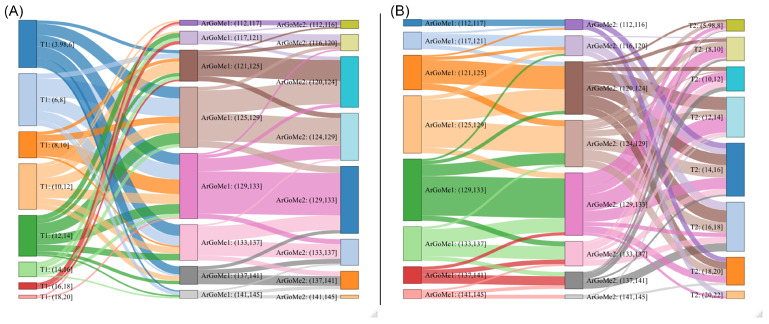
Trajectories of ArGoMe angle values during the growth process in 140 patients with Class III malocclusion, divided into eight classes of age. ArGoMe (Gonial angle) is the angle between the corpus and the ramus of the mandible. The Sankey diagram is usually used to indicate a data transfer in a process. In this data visualization, the width of the arrows is proportional to the number of feature flows. Sankey diagrams (**A**,**B**) draw attention to the transfer of values of the ArGoMe angle between two temporal acquisitions, T1 and T2. Plot (**A**) shows how the eight classes of age are distributed across the ArGoMe values at T1 and their evolution towards T2; plot (**B**) reveals how the ArGoMe values at T2 are distributed across the classes of age at T2. The Sankey diagram was obtained in ggplot2. Patient data were kindly offered by professors Lorenzo Franchi and James McNamara Jr.

**Figure 5 jpm-12-00957-f005:**
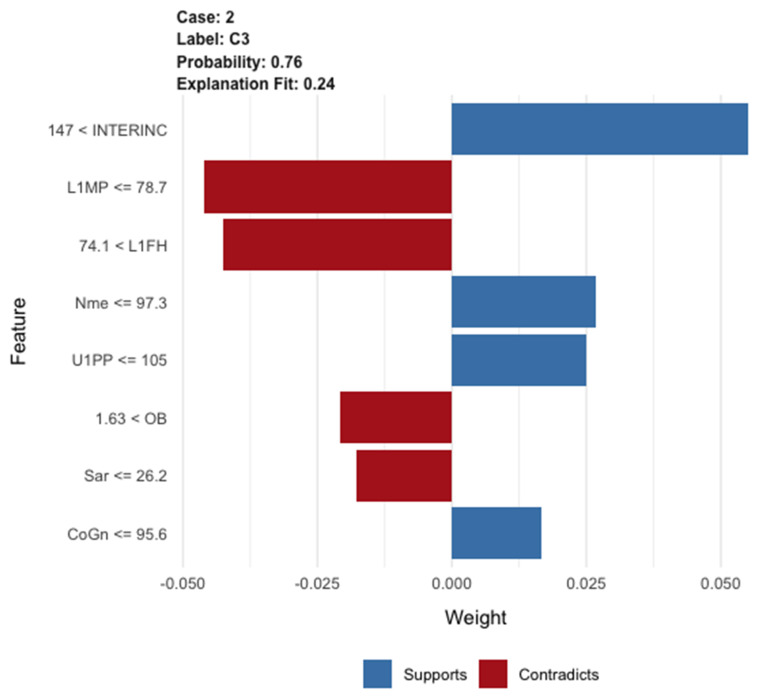
Trust in individual predictions is crucial when the model is used for treatment decisions. Using the LIME explanation procedure [54], we can explain the predictions of any classifier or regressor by approximating it locally with an interpretable model. The figure shows the cephalometric features of one Class III male patient with very bad maxillomandibular growth. The bar chart represents the importance of the most relevant cephalometric features that supported the prediction of increasing skeletal imbalance. The blue bars supported the predictions, whereas red bars contradicted them.

## Data Availability

Not applicable.

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
