# Peer review of "The Validity of Machine Learning Procedures in Orthodontics: What Is Still Missing?"

_jpm, 2022, doi:10.3390/jpm12060957_

Round 1

Reviewer 1 Report

Dear authors,

The manuscript looks good, but needs to be "polished" in order to be accepted for publishing.

  1. English revision should be performed.
  2. I suggest to modify the subtitles: for example subtitle 2: "Challenging interface between Machine Learning models and orthodontic features", subtitle 3: "How can orthodontic input could  be incorporated into the machine learning process?" and so on...
  3. The authors need to describe in detail how they have selected their papers included in the review, a number of those, how they have sorted, which websites were used and which method (for example: PRISMA) was used etc. This is a standard procedure for any review work.
  4. To classify this manuscript as a review paper on the topic mentioned the authors need to expand section 3 quite a bit (significantly). This section is the heart of the review. Authors are requested to invest more time in that. The authors can think of introducing a table, providing comparison, creating more subsections etc. Also, go into more detail about significant publications.

Author Response

  1. The language of the paper has been carefully revised.
  2. Subtitles have been modified according to the reviewer's suggestion
  3. We want to clarify that this paper aims to guide the attention of the interested readers to the pros and cons of AI in orthodontics and to offer a critical perspective view on the field. However, we have improved the selection of papers included in the manuscript, and we connected them better with the methods discussed. The papers have been selected among those found by using PubMed.
  4. We have significantly expanded section 3 by enriching the two appendices A end B to explain more thoroughly the issue related to the specificities of orthodontic craniofacial features and the related risks of misunderstanding.

Reviewer 2 Report

The purpose of the manuscript is to describe the conditions that need to be met for artificial intelligence to become useful in the orthodontic field. The authors have presented the potential of the application of machine learning in the orthodontic arena as well as posed the current limitations with future directions. From a perspective, the manuscript is nicely organized and well presented. The authors are suggested to work with the major/minor issues before the manuscript is suitable for publication in the “Journal of Personalized Medicine”.

Minor points:

  1. The authors are requested to do a better literature survey. They have not included any publications or reviews from 2021 or 2022. There are several (more than 10) articles published during those two years which are directly related to ML in orthodontics. Authors are requested to include the significant ones and evaluate their contributions too with proper citations. Also, authors need to give credit to those if they have come to a similar conclusion.
  2. In several places authors have mentioned the necessity of a large dataset with sufficient variation, Is there any such dataset currently available? What are the author's thoughts on a regulatory or central body to prepare such a dataset?
  3. Authors are requested to invest more space and time in feature optimization. Feature optimization can help the dentist to work with more relevant features as well as reduce the computational burden.

Minor points:

  1. Authors have mainly targeted the confidence and trust issues with ML in orthodontics. Accuracy, reliability, and consistency are also very significant criteria in diagnosis. Does the author know of any tool which is helping professionals in orthodontics with an accurate and consistent diagnosis?

Author Response

  1. We included additional recent results obtained in the orthodontic field using AI algorithms and correlated these data with recent bibliographic entries.
  2. For the benefit of more appropriate responses, we have emphasized the need to implement both quantity and quality of data by resorting to computational procedures described in Appendices A and B. Of course, we think that a regulatory or central body to prepare an adequate database. It has been included in the manuscript.
  3. This important aspect is also discussed in detail in Appendices A and B.
  4. Consistency of diagnosis can be improved by using proper clinical comparison background between the learning set and test set (adjusted for age, sex, etc.), resorting to selecting confounding variables, identifying dominant variables, and so on. These aspects are also discussed in Appendices A and B.

Round 2

Reviewer 1 Report

The manuscript is considerably improved. The authors have responded to all my requirements.Therefore I agree to the publication of this article.

Reviewer 2 Report

The authors have adequately answered all the queries.